# New Version of High-Damping PCB with Multi-Layered Viscous Lamina

**Tae-Yong Park** [1] , **Seok-Jin Shin** [2] **and Hyun-Ung Oh** [3,*]

1  Soletop Co., Ltd., 409 Expo-ro, Yuseong-gu, Daejeon 34051, Korea; typark@soletop.com
2  Drive Concept Development Team, Hyundai Motor Group, 150, Yonguso-ro, Namyang-eup, Hwaseong-si 18280, Korea; seokjin1126@hyundai.com
3  Space Technology Synthesis Laboratory, Department of Smart Vehicle System Engineering, Chosun University, 309, Pilmun-daero, Dong-gu, Gwangju City 61452, Korea
*  Correspondence: ohu129@chosun.ac.kr; Tel.: +82-62-230-7728; Fax: +82-62-230-7186

**Abstract:** In a previous study, a high-damping printed circuit board (PCB) implemented by multilayered viscoelastic acrylic tapes was investigated to increase the fatigue life of solder joints of electronic packages by vibration attenuation in a random vibration environment. However, the main drawback of this concept is its inability to mount electronic parts on the PCB surface area occupied by interlaminated layers. For the efficient spatial accommodation of electronics, this paper proposes a new version of a high-damping PCB with multilayered viscoelastic tapes interlaminated on a thin metal stiffener spaced from a PCB. Compared to the previous study, this concept ensures efficient utilization of the PCB area for mounting electronic parts as well as the vibration attenuation capability. Free vibration tests were performed at various temperatures to obtain the basic characteristics of the proposed PCB. The effectiveness of the proposed PCB was verified by random vibration fatigue tests of sample PCBs with various numbers of viscoelastic layers to compare the fatigue life of electronic packages.

**Keywords:** high-damping PCB; viscous lamina; fatigue life; random vibration; vibration attenuation

## 1. Introduction

In recent years, technological trends in various engineering fields such as automotive, defense, aeronautical, and space engineering have been rapidly changing owing to the industrial revolution. The emergence of autonomous transportation, unmanned aerial vehicles (UAVs), and large constellations of small satellites are several examples of these trends, which are changing commercial and military services [1–3]. Accordingly, these recent trends have driven advances in the functionality and performance of onboard electronics for various applications where the electronics are required to be more compact in volume and lighter in weight.

In the electronic engineering sector, efforts to reduce the mass and volume of electronics have mainly focused on the development and utilization of highly integrated electronic packages [4]. The application of these packages allows a printed circuit board (PCB) to be compact in volume, as they provide a high number of inputs/outputs and processing capabilities. Meanwhile, the mechanical engineering sector has focused on the minimization of the bulkiness of housing structures to integrate the PCB as the housing typically occupies more than 50% of the mass budget of the electronics [5,6]. Nevertheless, the housing structure shall provide sufficient strength and stiffness to prevent the failure of electronics during test and service periods. One of the major failure mechanisms of electronics is the fatigue failure of the solder joint of the electronic package owing to the stress induced by repetitive deflection of the PCB under vibration excitation. From a mechanical design point of view, a common and simple solution to prevent the vibration-driven failure of the solder joint was the application of a stiffener on the PCB to reduce the dynamic board

deflection. This structural reinforcement approach is the simplest way to ensure the fatigue life of solder joints; however, it involves an inevitable increase in the mass and volume of the electronics.

In previous studies, several other solutions such as underfill, corner stacking, and corner glue were proposed and studied to increase the fatigue life of solder joints [7–9]. In addition, some studies focused on an analytical approach to find the proper mounting location of the electronic package on the PCB in terms of solder stress minimization [10]. A more effective approach to ensure the solder fatigue life, in comparison with the afore-mentioned techniques [7–10], is to increase the vibration attenuation capability of the PCB. Representative examples include the applications of rubber mounts, potting materials, particle impact dampers (PIDs), and active vibration controllers [11–14]. Among these, PIDs have attracted considerable attention for electronics applications owing to their simplicity, low weight, and effective vibration attenuation performance. The PID is a form of damper that typically consists of a sealed container filled with particles such as metal balls. The vibration amplitude can be reduced by the kinetic energy generated by the collision of particles and friction between the particles in the container. To date, several studies have been performed to investigate various types of PIDs [13,15,16].

However, the PID has several drawbacks in its application to electronics. Typical PIDs require at least 20–30 mm of accommodation space in the out-of-plane direction of the PCB. This can lead to an increase in the distance between the PCBs, which increases the size of the housing structure of the electronics. Another potential issue is that multiple numbers of PIDs might be required to attenuate the PCB responses at various mounting locations of electronic packages if the PCB exhibits complex mode shapes caused by irregular and asymmetric board configurations. In addition, the noise generated by the repetitive impact of particles in a PID container under vibration might be disadvantageous for noise-sensitive applications.

Another vibration attenuation method called constrained layer damping (CLD) has also been previously investigated for electronic applications [17,18]. In general, a PCB applied with the CLD method consists of multiple constrained layers interlaminated with a viscous lamina on the PCB surface. This form is advantageous for attenuating the vibration response by the friction between the constrained layers and viscoelastic behavior of the viscous lamina material. In addition, the CLD method can overcome the aforementioned drawbacks of PID. Based on the concept of CLD, Park et al. [19] proposed a high-damping PCB for wedge-lock applications. Their results of basic characteristic tests and random vibration tests showed that the fatigue life of an electronic package could be increased by more than eight times using the proposed concept. However, even this concept has a drawback in terms of the spatial accommodation of electronic parts on the PCB because the viscous lamina is attached to the overall surface of the PCB, although some areas were opened to provide accessibility to the electronic parts.

To implement compact and lightweight electronics, more efficient spatial utilization of the PCB area is a necessary task. This is the starting point of this study. In the present study, we proposed a new version of a high-damping PCB to overcome the drawbacks of the previous concept [19]. The proposed new version of PCB has multiple viscous laminas interlaminated on a thin metal stiffener spaced from the PCB. This ensures efficient utilization of the PCB area for mounting the electronic parts because only some mechanical fixation parts occupy the PCB surface area, unlike the previous version [19]. To verify the new version of the high-damping PCB, free vibration tests at various temperature conditions were performed to obtain the basic characteristics of the PCB with viscous lamina. In addition, PCB samples with various numbers of viscous laminas were fabricated and exposed to a random vibration environment to compare the fatigue life of electronic packages.

## 2. Research Background

Figure 1 shows a representative example of the configuration of electronics. They typically consist of PCBs and housing structures to protect boards from external environ-

ments. The dynamic deflection of a PCB under vibration is inversely proportional to the square of the PCB eigenfrequency [7]. Therefore, a typical design approach to ensure the fatigue life of solder joints in electronic packages involves the implementation of additional metal stiffeners on the PCB. This solution has also been used for PCBs with wedge-lock applications [20,21]. The stiffener also acts as a conductive thermal path to dissipate the heat generated from the PCB to the housing structure.

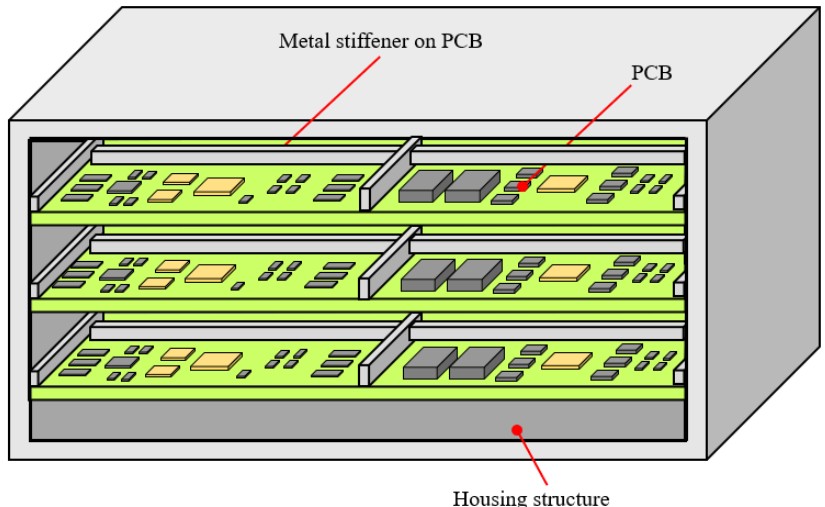

**Figure 1.** Example of configuration of electronics.

However, the above conventional design approach that focuses on increasing the board stiffness causes an increase in the mass and volume of electronics because the housing structure becomes bulky. The spatial accommodation of various electronic parts (packages, connectors, soldering cables, etc.) on the PCB surface also becomes less efficient because the area in contact with the stiffener cannot be utilized for mounting these parts. Another limitation is that in some cases the fatigue life of electronic packages might not increase as much, even if the board stiffness increases. This might be true considering that the board eigenfrequency is inversely proportional to the fatigue life, although the board deflection is the most dominant. For these reasons, the enhancement of the vibration attenuation capability of the PCB by increasing its damping property could be a much better solution than the conventional approach of using a stiffener, as long as there is no thermal dissipation problem in the electronics.

To solve the above technical issue, Park et al. [19] proposed a high-damping PCB with a multilayered viscous lamina as the configuration shown in Figure 2. Their proposed concept consisted of the FR-4 PCB, constrained layers made up of the same PCB material, and viscoelastic acrylic tapes to interlaminate the constrained layers on the bottom side of the board. The experimental test results showed that the high-damping PCB with five layers of stiffener achieved a damping ratio of 0.048, which was 3.67 times higher than that of the PCB without interlaminated layers. This resulted in more than 8.21 times increased fatigue life of the electronic package. However, the aforementioned problem associated with the spatial accommodation of electronic parts on the PCB has not been solved yet using this concept because the constrained layers were directly attached to the bottom surface of the board. Despite its effectiveness in enhancing the fatigue life of solder joints, the high-damping PCB needs to be improved in terms of the spatial efficiency of the PCB surface area. For this, efficient utilization of both the top and bottom sides of the PCB is possible while ensuring the damping capability. This would contribute to further minimization of the housing volume to implement compact and lightweight electronics.

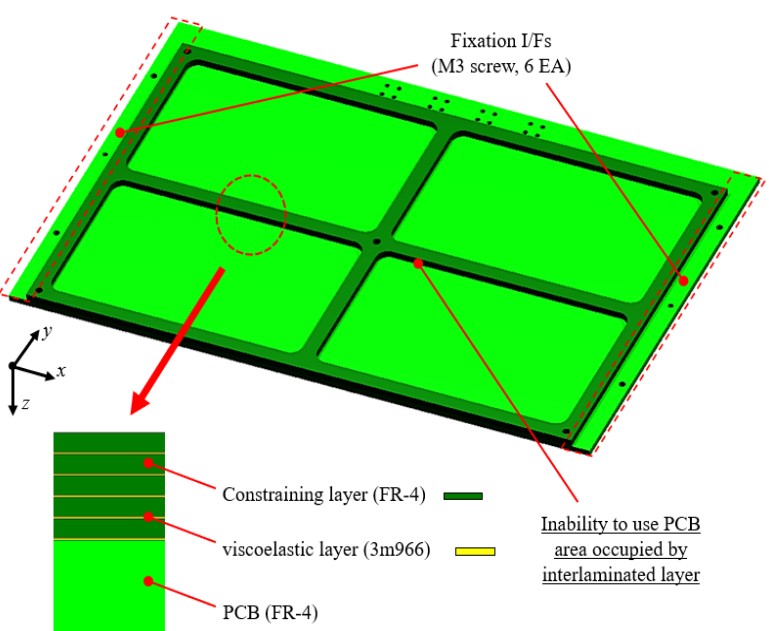

**Figure 2.** Configuration of previous version of high-damping PCB.

## 3. New Version of High-Damping PCB

In the present study, a new version of the high-damping PCB shown in Figure 3 was developed to overcome the limitations of the conventional PCB proposed in a previous study [19]. The new version consists of a PCB, constrained layers, and double-sided tapes with viscous lamina. The PCB is made of FR-4, and its dimensions are 243 mm × 160 mm × 1.2 mm. Constrained layers made of the same PCB material were attached to the thin aluminum stiffener using viscous lamina tape. The vibration attenuation capability of the proposed PCB subjected to out-of-plane excitation was mainly implemented by the dissipation of vibrational energy resulting from the shear deformation of the viscous lamina tapes with a high loss factor. The size of the stiffener with the viscous lamina tapes covers the entire area of the PCB; this can effectively attenuate the vibration caused by not only the first global bending modes but also the local modes at higher frequencies. This ensures the vibration attenuation performance of PCBs having complex mode shapes caused by the irregular and asymmetric board configurations.

The stiffener has 15 mechanical standoff interfaces to integrate the stiffener on the PCB, which allows the stiffener to be spaced 3 mm away from the PCB. This configuration enables the utilization of the surface areas on both sides of the PCB for mounting the electronic parts. This is the main advantage that differentiates this model from the previous version [19]. In addition, the PCB thickness required to meet the board stiffness requirement could be minimized as it was stiffened by the thin aluminum stiffener. Furthermore, a disassembly process of the entire stiffener with constrained layers can be easily performed if the rework of some electronic parts is required. This is an additional advantage in comparison with the previous version [19], which had difficulty in disassembly as the constrained layers were directly attached to the PCB.

Figure 4 shows the assembly process of the proposed PCB for secure and uniform adhesion of the viscous lamina tape to the stiffener. First, individual constrained layers were attached to 3M 966 tape (process 1–3). An aluminum stiffener was placed on the bottom integration jig, and the constrained layers were stacked on the stiffener (process 4–5). Four guide interfaces integrated with the jig were used for the proper attachment of each layer (process 6). After completion of the attachment process, the top integration jig was placed on the stiffener with viscous lamina layers, and uniform pressure was applied to it by fastening M3 screws at various locations of the jig (process 7–8). The torque for M3 screws applied to the specimens were 1.2 N-m. The pressure was maintained for 72 h to

cure the tape. Finally, the PCB was integrated with the interlaminated stiffener by fastening with M2.5 low-profile head screws (process 9).

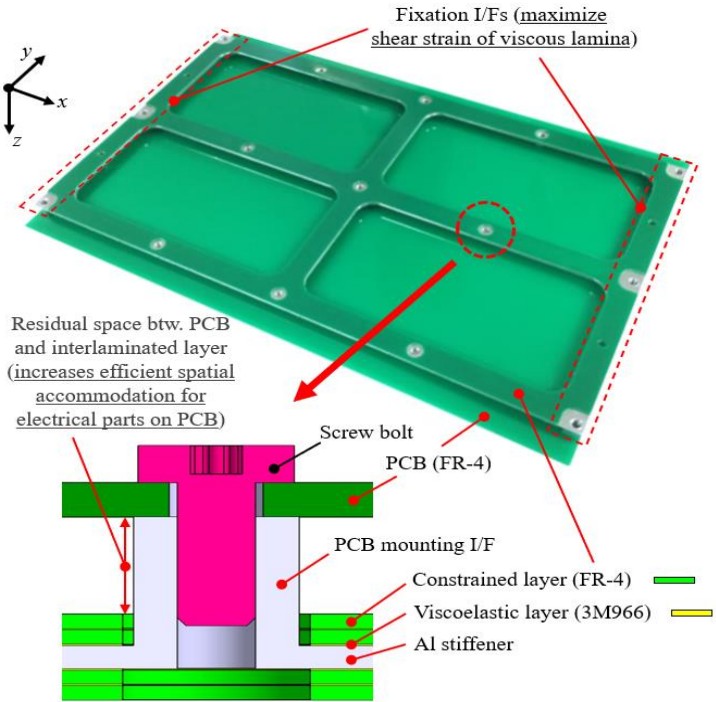

**Figure 3.** Configuration of new version of high-damping PCB.

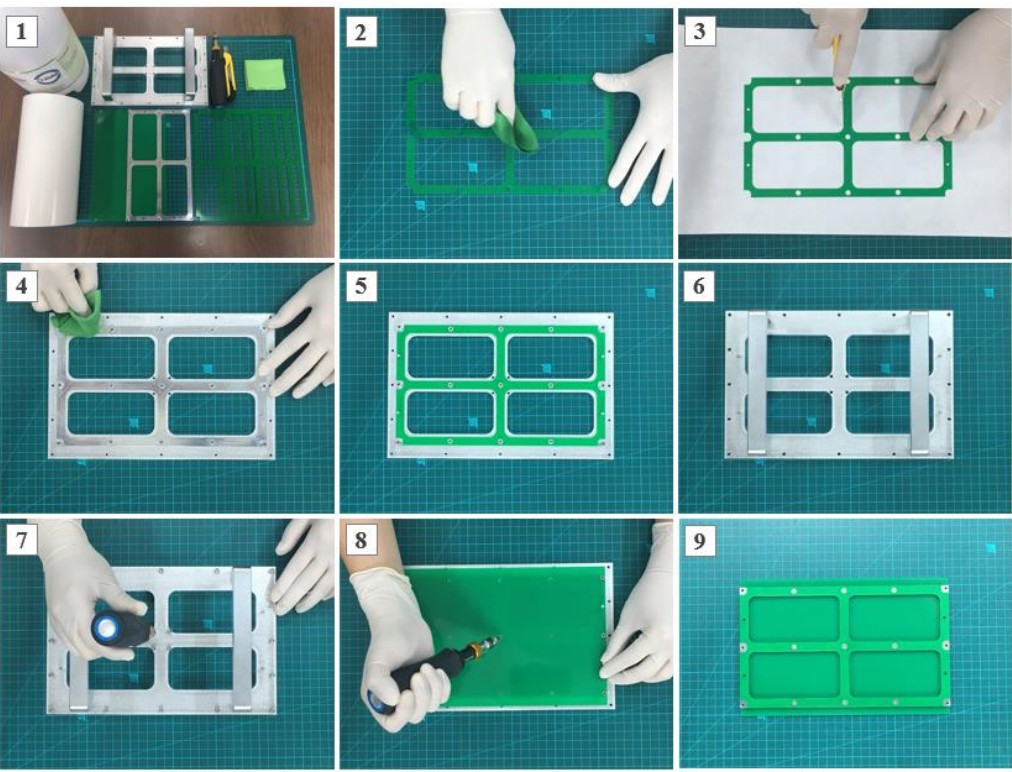

1) Prepare sample PCB, Al stiffener, constrained layer, 3M966 tape, cutter, cleaner, isopropyl alcohol (IPA). 2) Clean constrained layer using cleaner and IPA. 3) Attach constrained layer on 3M966 tape and cut. 4) Clean Al stiffener on bottom integration jig. 5) Remove protection layer of 3M966 tape attached on constrained layer and attach it on Al stiffener. [Repeat process 2) ~ 5) for each layer] 6) Place top integration jig on the sample. 7) Apply torque on top integration jig using M3 screws (torque: 1.2 N-m). 8) Integrate PCB and Al stiffener with interlaminated layers. 9) Assembly process completed.

**Figure 4.** Assembly process of new version of high-damping PCB.

The specifications of the new version of the high-damping PCB are listed in Table 1. In this study, three PCB sample assemblies with various numbers of interlaminated layers of viscous lamina were fabricated to compare their dynamic responses. Case 1 corresponds to the PCB without viscous lamina layers; however, the stiffener is integrated on the board. Cases 2 and 3 correspond to the PCB with two and four layers of viscous lamina, respectively. 3M 966 double-sided viscous lamina tape [22] was used for the sample PCB assemblies owing to its applicability in various applications including space applications. The sufficient adhesion strength of the tape for ensuring mechanical safety under a severe vibration environment was demonstrated in a previous study [19]. Even the PCB assembly in case 3 had a total thickness of only 8.7 mm, which might be two to three times more compact in the out-of-plane dimension as compared with a PCB having conventional PIDs. This indicates that the proposed concept of PCB is effective for the implementation of compact and lightweight electronics.

**Table 1.** Specifications of new version of high-damping PCB.

| Item | Specification | | |
|:---:|:---:|:---:|:---:|
| Case | 1 | 2 | 3 |
| Material | FR-4 (PCB, Constrained Layer) Aluminum 6061-T6 (Stiffener) | | |
| Young's Modulus | 18.73 GPa (FR-4) 68.9 GPa (Aluminum 6061-T6) | | |
| PCB Dimension (mm) | 243 × 160 × 1.2 | | |
| No. of Layers | 0 | 2 | 4 |
| Total Thickness (mm) | 7.7 | 8.2 | 8.7 |
| Mass (g) | 123 | 142 | 158 |

## 4. Basic Characteristics

To investigate the basic characteristics of the proposed new version of the high-damping PCB, free vibration tests of the PCB samples with various numbers of interlaminated layers were performed. Figure 5 shows the free vibration test setup. The specimen assembly integrated on the fixation jig were installed in the thermal cycling (TC) chamber (ST-200CO, SALT Co., Incheon, South Korea). The TC chamber is capable of implementing temperature ranging from −20 °C to 60 °C with a temperature change rate of 1 °C/min. The accelerometer sensor with a mass of 3 g was attached at the location adjacent to the center of the PCB to measure the free vibration response of the specimen, which was caused by applying a tapping force on the board. The reading frequency of the acceleration was 2000 Hz, which was sufficient to observe the first resonant frequency of the board. The temperature condition was achieved by chamber control. The time histories of the acceleration responses measured at the centers of samples 1, 2, and 3 at room temperature condition (20 °C) are depicted in Figure 6. These time histories show that the vibration damping performance increases as the number of interlaminated layers increases. The first eigenfrequency and damping ratio of each PCB sample derived from the time histories are summarized in Table 2. The damping ratio of the case 3 sample is 0.061, which is 5.1 times higher than that of case 1. This improvement was caused by the more effective viscoelastic behavior of the lamina tapes resulting from the increased shear strain in the interlaminated layers.

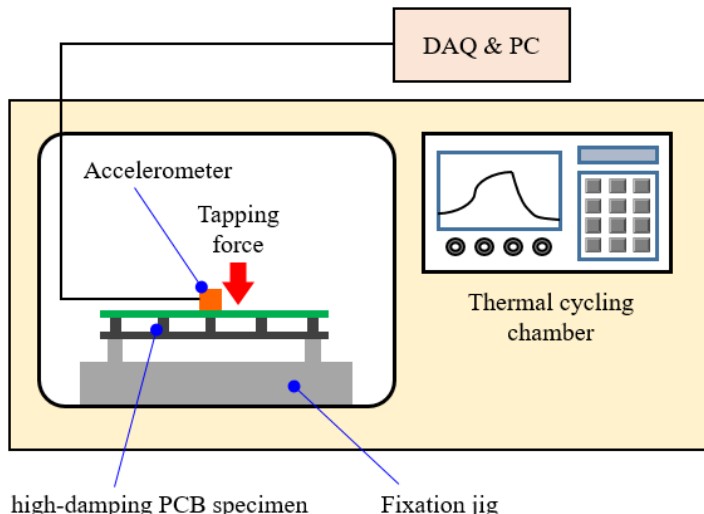

**Figure 5.** Free-vibration test setup for PCB specimen in thermal cycling chamber.

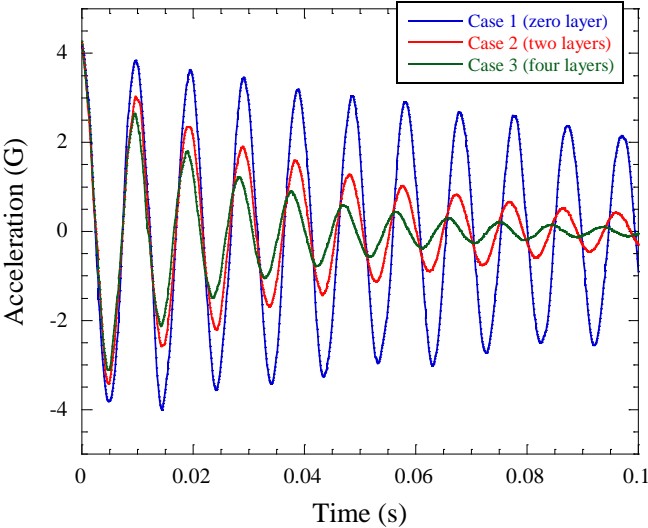

**Figure 6.** Time histories of measured acceleration responses of cases 1, 2 and 3 samples.

**Table 2.** First eigenfrequency and damping ratio of cases 1, 2 and 3 samples.

| Case | 1st Eigenfrequency (Hz) | Damping Ratio | Remarks |
|------|------------------------|---------------|---------|
| 1 | 102.5 | 0.012 | - |
| 2 | 105.0 | 0.033 | Damping ratio: 2.8 times increased |
| 3 | 106.6 | 0.061 | Damping ratio: 5.1 times increased |

As reported in a previous study [19], viscous lamina tape exhibits a temperature-dependent variation in basic characteristics. Therefore, the basic characteristics of the proposed PCB were also examined through additional free vibration tests at various temperatures ranging from −20 to 60 °C. The temperature conditions were set using a thermal chamber. Figure 7 shows the damping ratios of the PCB samples obtained under different temperature conditions. The figure also includes the results shown in Figure 6 obtained at room temperature. Cases 2 and 3 PCB samples with viscous lamina layers on the stiffener exhibited a higher damping ratio than that of case 1 without layers. The board damping ratio increased as the temperature increased beyond room temperature. This was caused by the increased energy dissipation under vibration owing to the shear strain until the glass transition temperature was reached. On the other hand, the viscous

lamina tape entered the glass transition phase at low temperatures. This caused a decrease in the damping ratio because the tape exhibited less viscoelastic behavior and more closely resembled an elastic material. Even so, a damping ratio that was at least 2.9 times that of case 1 was observed in the case 3 sample. The behaviors observed from the samples were also observed in a previous study [19]. These test results confirm that the proposed PCB ensures vibration damping performance under various temperature conditions.

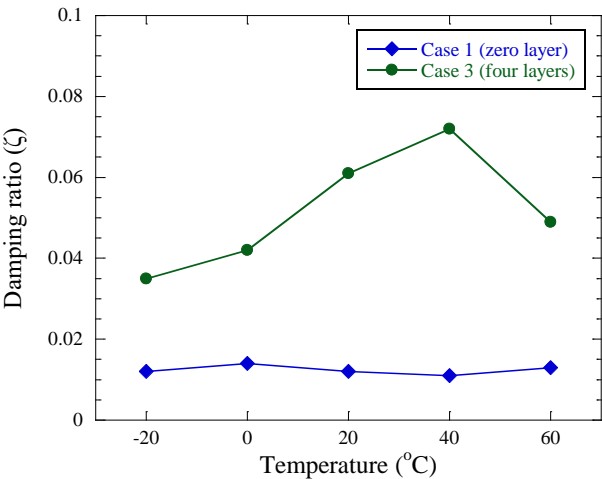

**Figure 7.** Measured damping ratio of cases 1 and 3 samples.

Random vibration tests of the PCB samples were performed at room temperature to confirm the vibration attenuation performance of the proposed PCB. Figure 8 shows an example of the random vibration test setup for the case 1 PCB sample. An accelerometer sensor was attached near the center of the board to measure the PCB response. A sensor to control the input acceleration was attached to the test jig. The sample was exposed to random vibration along the *z*-axis because the out-of-plane direction of the board is the most critical for the solder joint fatigue life of the electronic package. Table 3 shows the qualification level specification of the random vibration input, which has been widely applied for testing spaceborne instruments.

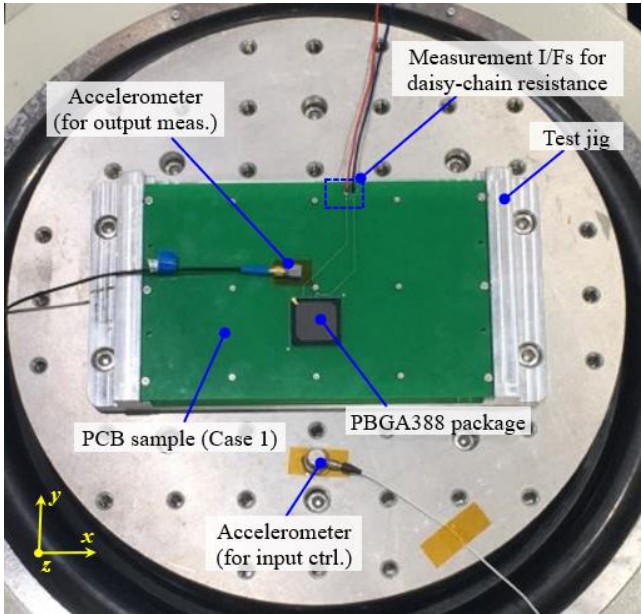

**Figure 8.** Example of random vibration test setup for case 1 PCB sample.

**Table 3.** Random vibration test specification.

| Frequency (Hz) | PSD Acceleration (g²/Hz) |
|---|---|
| 20 | 0.091 |
| 60 | 0.273 |
| 1000 | 0.273 |
| 2000 | 0.069 |
| Overall (full level (0 dB)) | 20 grms |

The power spectral density (PSD) acceleration profiles obtained from the PCB samples are plotted in Figure 9, and Table 4 summarizes the root mean square (RMS) acceleration and maximum displacement values of the PCB samples derived from the PSD profiles. It can be observed that the overall PSD response over the excitation frequency is reduced as the number of viscous lamina layers increases. In particular, case 3 with four viscous lamina layers showed 25.78 grms and the maximum displacement (3-sigma value) was 0.69 mm. This displacement value is approximately 2.3 times lower than that of case 1 without viscous lamina layers. These results indicate that the multilayered viscous lamina tapes are effective in attenuating the random vibration owing to their viscoelastic behavior and the friction between the interlaminated layers on the stiffener.

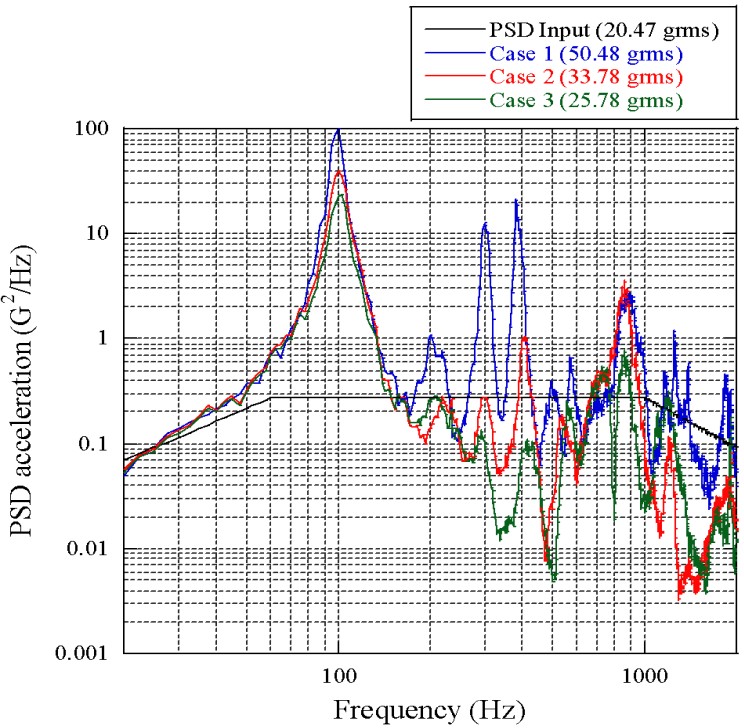

**Figure 9.** PSD acceleration profiles measured from cases 1, 2 and 3 samples.

**Table 4.** RMS acceleration and maximum displacement of cases 1, 2 and 3 samples.

| Case | RMS Acceleration (grms) | Max. Displacement (mm) | Remark |
|---|---|---|---|
| 1 | 50.48 | 1.58 | - |
| 2 | 33.78 | 0.99 | - grms 1.49 times reduced<br>- Max. disp. 1.60 times reduced |
| 3 | 25.78 | 0.69 | - grms 1.96 times reduced<br>- Max. disp. 2.29 times reduced |

## 5. Fatigue Life Tests

To verify the enhancement in the fatigue life of solder joints for electronic packaging in a random vibration environment by applying the proposed concept of PCB, cases 1–3 PCB samples with electronic packages were fabricated, and their fatigue life tests were performed. An additional objective of this test was to validate the structural safety against the delamination of viscous lamina layers caused by the repetitive shear and peeling stress on the tape attachment interface under random vibration, although this was also addressed in a previous study [19]. This is because, as described above, these layers are attached to the aluminum stiffener, and therefore the adhesion surface material is different from the previous version of the high-damping PCB.

The fatigue life test setup and vibration excitation axis are the same as those of the random vibration test shown in Figure 8. A 388-pin plastic ball grid array (PBGA-388) package was mounted on the location adjacent to the center of the PCB sample. Eutectic solder (Sn63-Pb37) was used to mount the package. The specifications of the PBGA-388 package are listed in Table 5. As the test objective was to compare the time to failure between sample cases with various numbers of viscous lamina layers, one PCB sample per case was fabricated as in a previous study [19].

**Table 5.** Specifications of PBGA388 package.

| Item | Specification |
| --- | --- |
| Manufacturer | Topline Co. Ltd., Milledgeville, GA, USA |
| Configuration |  |
| Solder ball | - Solder material: Sn63-Pb37<br>- Dimension (mm): 0.45 × 0.7 (height × maximum ball diameter)<br>- Solder pitch (mm): 1.27<br>- No. of solder balls (EA): 388 |
| Package | - Package material: BT substrate with mold encapsulation<br>- Dimension (mm): 35 × 35 × 1.65 (after reflow soldering on PCB)<br>- Weight (grams): 5.0 (incl. solder balls) |

The time to failure of the solder joint was measured using an in situ resistance monitoring method during vibration excitation. For ease of resistance monitoring, the PBGA-388 package selected in this study had a daisy-chain circuit, which is widely used for mechanical testing. In other words, all solder joints were connected in series when mounted on the PCB. Resistance monitoring and logging were performed using a digital multi-meter (Keithley Instruments Inc., Solon, OH, USA, DAQ6510) at a speed of 1.7 samples/s. The failure criterion of the package, which is the crack occurrence of the solder joint, was defined as when a 20% increased resistance over the initial value was detected for five consecutive readings in accordance with the IPC-9701A standard [22]. To investigate the occurrence of delamination in the interlaminated layers or any other mechanical failure after completion of the test, an accelerometer sensor was installed near the center of the PCB. The relevant failure criterion is defined as when the PCB sample exhibits a variation in the first eigenfrequency of more than 5% [23].

Figure 10 shows the time histories of the daisy-chain resistance of the sample packages for cases 1, 2, and 3. The increase in resistance value of the case 1 sample without viscous

lamina layers reached the failure criterion at 4.77 h. The case 2 and 3 samples reached the failure criterion at 12.19 h and 14.99 h, respectively. Consequently, a 3.14-times increase in time to failure was achieved by the case 3 sample with four viscous lamina layers as compared to the case 1 sample without layers. These observations indicate that the proposed concept of PCB effectively enhances the fatigue life of electronic packages in a random vibration environment owing to the improved damping performance of multilayered viscous lamina tapes applied on the stiffener.

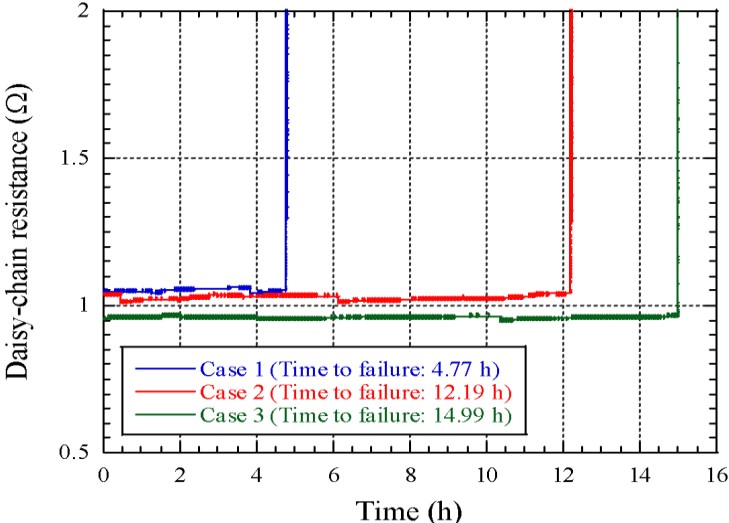

**Figure 10.** Time histories of daisy-chain resistance of cases 1, 2 and 3 samples.

Figure 11 shows the low-level sine sweep results for resonance survey of case 2 and 3 samples before and after the fatigue life test. The first eigenfrequencies measured from Figure 9 are summarized in Table 6. The tested samples did not reach the failure criterion, as the frequency variations were less than 1%. In addition, visual inspection of the tested samples was performed. Figure 12a,b show the representative inspection results—that is, optical microscope images of the multilayered viscous lamina layers for cases 2 and 3 taken at the sidereal edges of these samples, respectively. No mechanical failures or delamination were observed in the samples with viscous lamina. These observations confirmed the structural safety of the proposed PCB in a random vibration environment.

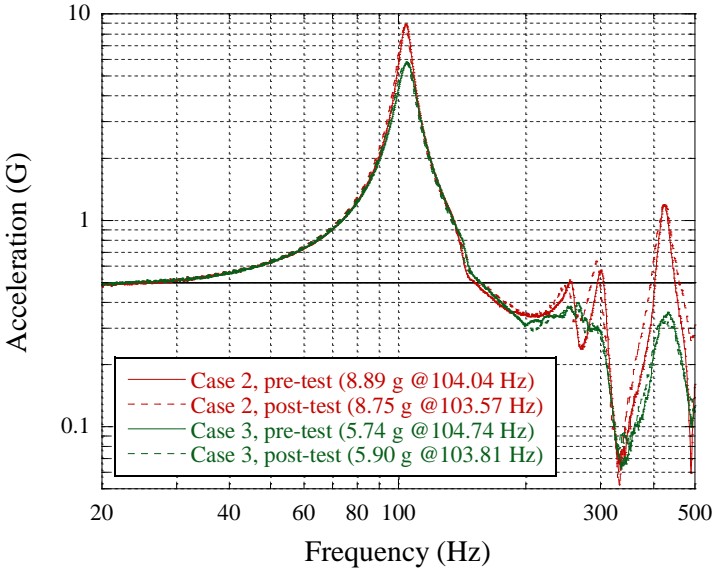

**Figure 11.** Resonance survey results on cases 2 and 3 samples.

**Table 6.** Measured first eigenfrequency of cases 2 and 3 samples before and after fatigue life test.

| Case | Vibration Exposure Time (h) | Pre-Test (Hz) | Post-Test (Hz) | Difference btw. Pre- & Post-Test (%) |
|------|-----------------------------|---------------|----------------|--------------------------------------|
| 2 | 12.19 | 104.04 | 103.57 | 0.45 |
| 3 | 14.99 | 104.74 | 103.81 | 0.88 |

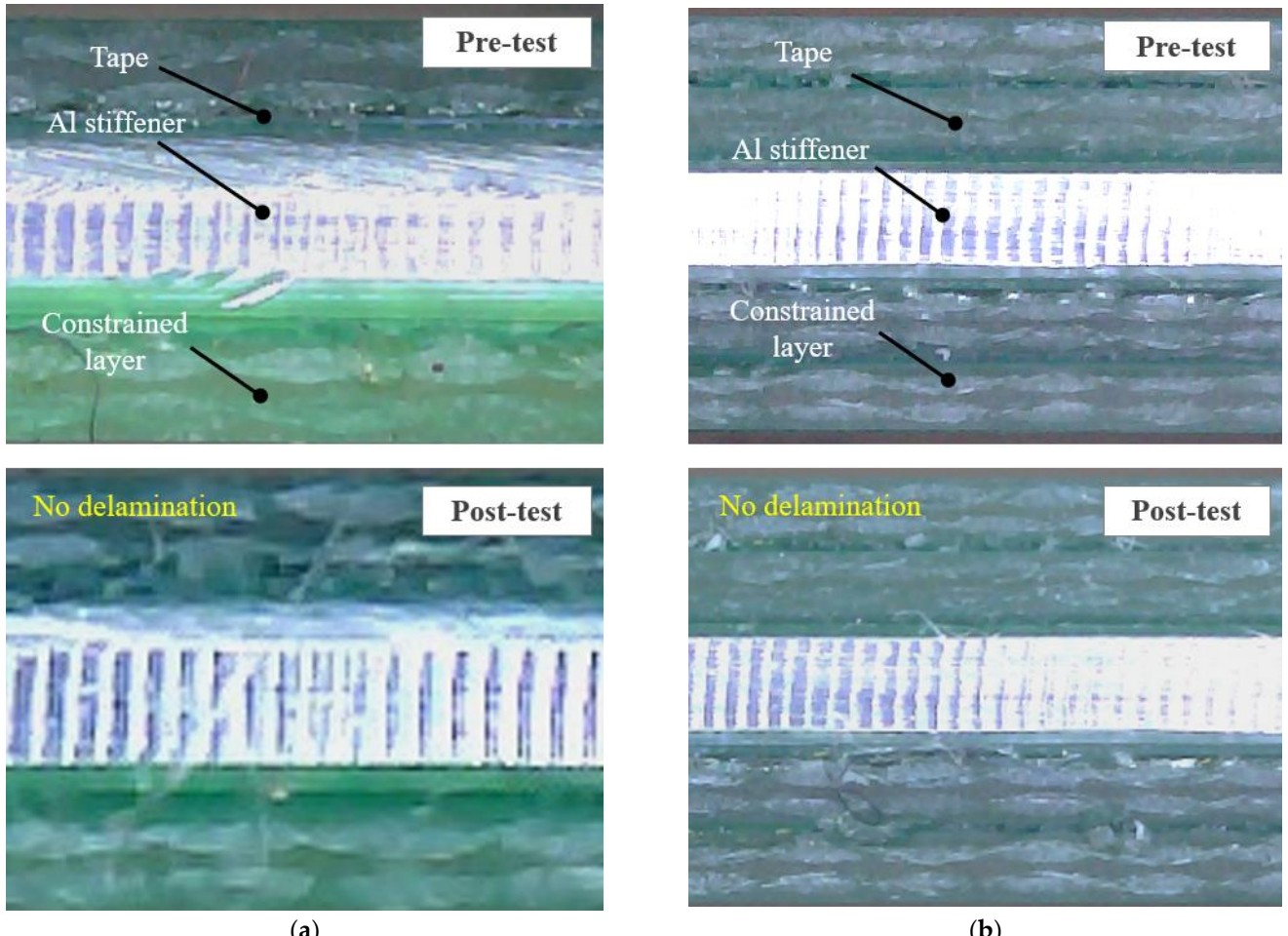

(**a**)             (**b**)

**Figure 12.** Optical microphotographs of interlaminated layers of (**a**) cases 2 and (**b**) 3 samples.

## 6. Conclusions

Herein, a new version of a high-damping PCB with viscous lamina tapes is proposed to overcome the drawbacks of the previous version in terms of the spatial accommodation efficiency of electronics. The main advantage of the new version in comparison with the previous version is that multilayered viscous lamina tapes are interlaminated on the thin metal stiffener spaced from the PCB. This enables the utilization of an increased area of PCB to accommodate electronic parts such as components, connectors, and harnesses, which leads to the minimization of the mass and volume of electronics. In addition, it provides increased fatigue life on solder joints of electronic packages by implementing vibration attenuation capability on the PCB exposed to a random vibration environment. To validate the proposed new version of PCB, free vibration tests were performed at various temperature conditions, and the results indicated that the damping capability was higher than that of a PCB without interlaminated layers. For the fatigue life test, PCB samples with various numbers of interlaminated viscous lamina tape layers were fabricated and exposed to a random vibration environment until the failure of the electronic packages. The test results indicated that the proposed concept effectively increased the fatigue life of

electronic packages by reducing the dynamic deflection of the PCB under random vibration. In the near future, further studies on high-damping PCBs will be conducted for actual applications in electronics.

**Author Contributions:** Conceptualization, H.-U.O.; validation, S.-J.S.; investigation, S.-J.S. and T.-Y.P.; data curation, S.-J.S. and T.-Y.P.; writing—original draft preparation, T.-Y.P.; writing—review and editing, T.-Y.P.; supervision, H.-U.O.; project administration, H.-U.O.; funding acquisition, H.-U.O. All authors have read and agreed to the published version of the manuscript.

**Funding:** This research was funded by Ministry of Education (MoE) of Korea, grant number NRF-2018R1D1A1B05047385. The APC was funded by Soletop Co. Ltd.

**Data Availability Statement:** The data used to support the findings of this study are available from the corresponding author upon request.

**Acknowledgments:** This research was supported by the National Research Foundation of Korea (NRF, NRF-2018R1D1A1B05047385).

**Conflicts of Interest:** The authors declare that they have no conflict of interest.

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
