# Peer review of "New Version of High-Damping PCB with Multi-Layered Viscous Lamina"

_aerospace, doi:10.3390/aerospace8080202_

Round 1

Reviewer 1 Report

Thank you for such interesting paper.

Please provide torque applied to screws.

Please define how figure 5 is obtained. 4G at 100Hz means a displacement of 0.4mm approximately. Do you apply an initial central displacement and monitor acceleration? What is the weight of accelerometer? Did you consider the use of readings without adding mass ass for example https://doi.org/10.1177/1077546314547728 ?

How is the reading frequency?

Could you include in figure 11 Case 1 images? It would be interesting to know how failure is produced with zero layers.

Table 1 is very useful. Would it be possible to add Young modulus for each layer? Al 69GPa...

Why layer 

Author Response

Thank you for taking time to review our manuscript.

Our responses can be found on the attached file. Please see those files and evaluate that the responses are appropriate.

Thank you.

Reviewer 2 Report

The article presents an experimental study of the vibration resistance of the augmented PCB plates to reduce vibrations and consequently prolong the fatigue life of solder joints. Basically the article looks fine, but a description of the experimental set-up should be improved as follows:

1.) Page 5, lines 166-175: please add reference numbers 1, 2, 3 etc. from Figure 4 to the text.

2.) Page 6, lines 193-196: a description of the test is too brief. Make it more clear. Additional figures/sketches would be most welcomed. Namely, how was the so called free vibration test performed?

3.) Page 7, lines 206-210: please, explain more thoroughly, how was the vibration test performed in the chamber. What kind of chamber was applied?

Additional comments:

- Please correct the square brackets at refernces 20, 21 and 22.

Author Response

The responses were described in the attached file. Thank you for your time to review our manuscript.
